# Association of pulmonary artery bifurcation angle shift with contralateral stenosis after post-arterial switch in the pediatric transposition of great artery patients

Panupong Seripanu[1], Tanop Srisuwan[2], Yupada Pongprot[3], Rekwan Sittiwangkul[3], Thanaporn Phanacharoensawad[1], Pakpoom Wongyikul[4], Phichayut Phinyo[4], Kwannapas Saengsin[3]*

1 Department of Pediatrics, Faculty of Medicine, Chiang Mai University, Chiang Mai, Thailand, 2 Department of Radiology, Faculty of Medicine, Chiang Mai University, Chiang Mai, Thailand, 3 Division of Cardiology, Department of Pediatrics, Faculty of Medicine, Chiang Mai University, Chiang Mai, Thailand, 4 Center for Clinical Epidemiology and Clinical Statistics, Department of Family Medicine, Faculty of Medicine, Chiang Mai University, Chiang Mai, Thailand

* Kwannapas_09@hotmail.com

## Abstract

### Background

The arterial switch operation (ASO) is the standard surgical treatment for transposition of the great arteries (TGA). Postoperative complications such as branch pulmonary artery (PA) stenosis are commonly observed. This study aimed to investigate the possible potential anatomical mechanisms contributing to isolated branch PA stenosis using cardiac computed tomography (CCT).

### Methods

A retrospective, single-center study was conducted on pediatric patients under 18 years of age diagnosed with TGA, with or without ventricular septal defect (VSD), who underwent ASO and cardiac CCT between January 2004 and October 2022. Baseline characteristics, echocardiographic data, and CCT findings were compared between patients with and without isolated branch PA stenosis. Special attention was given to the angle between the PA bifurcation and the ascending aorta (AAO).

### Results

Among 30 patients enrolled, 46.67% (14/30) had isolated branch PA stenosis, while 53.33% (16/30) had no stenosis. Baseline and echocardiographic variables showed no significant differences between groups, except for the PA bifurcation angle relative to the AAO. Patients with isolated right PA (RPA) stenosis demonstrated significantly greater leftward angulation (median angle: 25° [IQR: 24, 30]) compared to the

**Data availability statement:** The data underlying this study cannot be made publicly available due to patient confidentiality restrictions. However, the data will be made available to qualified researchers upon reasonable request. Data requests may be directed to the Research Ethics Committee of the Faculty of Medicine, Chiang Mai University which is responsible for oversight of data governance and access. Please contact: Research Ethics Committee, Faculty of Medicine, Chiang Mai University Address: Rianroum Buiding, 3rd floor Faculty of Medicine, Chiang Mai University 110 Intavaroros Road, Tambon Sriphum, Amphoe Muang, Chiang Mai Thailand 50200 Business Tel.: No. +6653936643, +6653936641, +6653935279 ext. 22-23 Email: research.med.cmu@gmail.com, researchmed@cmu.ac.th. This committee is independent of the study authors and is authorized to review and approve data access requests in accordance with institutional policies and ethical guidelines.

**Funding:** The author(s) received no specific funding for this work.

**Competing interests:** The authors have declared that no competing interests exist.

**Abbreviation:** AAO, Ascending Aorta; ASO, Arterial Switch Operation; ASD, Atrial Septum Defect; BSA, Body Surface Area; CCT, Cardiac Computerized Tomography; CMR, Cardiovascular Magnetic Resonance; CTA, Computer Tomography Angiography; GDM, Gestational Diabetes Mellitus; IQR, Interquartile Range; LPA, Left Pulmonary Artery; IVS, Intact Ventricular Septum; LV, Left Ventricle; LVEF, Left Ventricular Ejection Fraction; PA, Pulmonary Artery; PDA, Patent Ductus Arteriosus; PEG1, Prostaglandin E1; RPA, Right Pulmonary Artery; RV, Right Ventricle; SD, Standard Deviation; TGA, Transposition of the Great Arteries; VSD, Ventricular Septal Defect

no-stenosis group (median angle: −2° [IQR: −8, 6.5]). Similarly, isolated left PA (LPA) stenosis was associated with greater rightward angulation (median angle: −22° [IQR: −32, −20]) compared to the no-stenosis group (P = 0.042).

## Conclusion

Alterations in the PA bifurcation angle relative to the AAO may contribute to the development of isolated branch PA stenosis following ASO. Preoperative planning to optimize the PA bifurcation angle to within ±20 degrees may help reduce the risk of postoperative stenosis. A larger study incorporating advanced cardiac imaging into routine ASO follow-up is warranted.

## Introduction

Transposition of the great arteries (TGA) is the most common cyanotic congenital heart disease presenting in the neonatal period [1]. The prevalence of TGA is estimated to be between 20 and 30 per 100,000 live births, based on population-based studies conducted in Europe and North America from 2000 to 2010 [2]. TGA is characterized by the aorta arising from the morphologic right ventricle (RV) and the pulmonary artery (PA) arising from the morphologic left ventricle (LV) [3]. Thirty percent of these infants will die in the first week of life, and 90% within the first year without treatment [4]. The operation of choice is the arterial switch operation (ASO), which reestablishes normal blood flow and proper alignment of the ventricles and great arteries. It involves transecting the aorta and pulmonary arteries above the sinuses and attaching them to their correct ventricles. The pulmonary arteries are moved anteriorly to the aorta using the LeCompte maneuver, while the coronary arteries are detached and implanted onto the neo-aorta [5]. Although the mortality rates have decreased with improved surgical techniques, short-term and long-term complications after an ASO are still observed, including stenosis of the pulmonary artery (PA) side branches, right ventricular outflow tract (RVOT) obstruction, neo-aortic valve insufficiency, dilatation of the aortic root, and obstruction of the coronary arteries [6–8]. The stenosis of the branch PAs, particularly supravalvular pulmonary stenosis, occurs mainly after an ASO. It is one of the most frequent reasons for reoperation and reintervention. Late pulmonary stenosis occurs in 5% to 15% of patients who have undergone an ASO [9]. The mechanism of branch stenosis is still unclear. According to the previous study, geometric anatomical factors contributing to branch pulmonary stenosis likely have multiple causes, with post-ASO branch pulmonary stenosis potentially arising from a lengthened and tensioned pulmonary artery pathway [10].

Cardiac computerized tomography (CCT) provides information on the morphology of the cardiovascular structures, including branch PA stenosis, ascending aorta dilatation, and coronary artery anatomy [11]. A previous study showed neo-pulmonary to neo-aortic geometry and post-operative compression of the left pulmonary artery (LPA) by an enlarged aorta impact on LPA size and perfusion of the left lung [10].

We aim to explore the possible mechanisms of branch PA stenosis using CCT to evaluate anatomy structures, including the stretched vessels of the pulmonary arteries and dilated ascending aorta. We hypothesize the new geometric anatomy orientations, including the angle between AAO and PA bifurcation after surgery are the risk factors of branch PA stenosis. It would then be possible to address the geometric structure intra-operatively and provide more comprehensive post-operative surveillance in patients with TGA after ASO surgery.

## Materials and methods

### Study population

We conducted a retrospective observational design in a single-center study. We reviewed data on consecutive pediatric TGA patients who underwent CCT study following ASO surgery including the LeCompte maneuver between 1/01/2004–31/10/2022. The institutional review board of the faculty of medicine, Chiang Mai University, Chiang Mai, Thailand, approved on 16/11/2022.

### Ethics approval process

This is a retrospective analysis of TGA patients who presented with isolated branch pulmonary artery stenosis. The study protocol was approved by the Ethics Committee of the Faculty of Medicine, Chiang Mai University on 16/11/2022 (approval no. 411/2022, date of approval 16/11/2022 and date of expiration 15/11/2023). The Principal Investigators had access to all information that could identify participants during or after data collection. Data were evaluated and collected for research purposes from 1/12/2022–15/11/2023. The investigations were carried out in accordance with the Declaration of Helsinki (2024), which outlines ethical principles for medical research involving human subjects, including research using anonymous human data.

**Inclusion and exclusion criteria.** The inclusion criteria included TGA pediatric patients, including TGA with intact ventricular septum (TGA/IVS) and TGA with ventricular septal defect (TGA/VSD) under the age of 18. We excluded TGA with complex congenital heart disease, such as double outlet of the right ventricle (DORV) and interrupted aortic arch (IAA). Studies with poor image quality and exams performed following PA reinterventions were excluded (Fig 1).

**Cardiac computerized tomography (CCT) protocol.** All patients received CT scans using one of two types of advanced CT systems: either dual-source CT systems (Somatom Force or Somatom Definition from Siemens) or dual-layered detector CT (iQon Spectral CT from Philips). For pediatric patients, special care was taken to optimize radiation dose.

In all cases, ECG-gated CT angiography (CTA) of the heart was performed with intravenous contrast media. The CTA protocol typically used either prospective high-pitch acquisition or retrospective acquisition with dose modulation. The choice depended on whether the focus was on assessing cardiac function or evaluating detailed cardiac anatomy.

Images were reconstructed into thin 0.6 mm slices in the transaxial plane. Post-processing of these images was done using Syngo.via software from Siemens.

**Neo-pulmonary to neo-aortic angle (Fig 2).** This is the angle between a line through the centers of the neoaortic and neopulmonary roots and an imaginary line drawn from the spinous process toward the anterior spinal border convexity.

**Pulmonary artery bifurcation to ascending aortic angle (Fig 2).** Pulmonary artery bifurcation to ascending aortic angle is defined as the angle between two lines: one line extends from the center of the ascending aorta to the pulmonary artery bifurcation, while the other is a reference line drawn from the spinous process to the anterior spinal border convexity. To measure this angle, the axial images are first scrolled through to locate the slice showing the tip of the reverse V-shaped pulmonary arterial bifurcation. A line is then drawn from the tip of the pulmonary arterial bifurcation to the center of the ascending aorta. The reference vertical line is drawn from the tip of the spinous process to the anterior most point of the spinal body convexity. This reference line should pass through the center of the spinal canal and

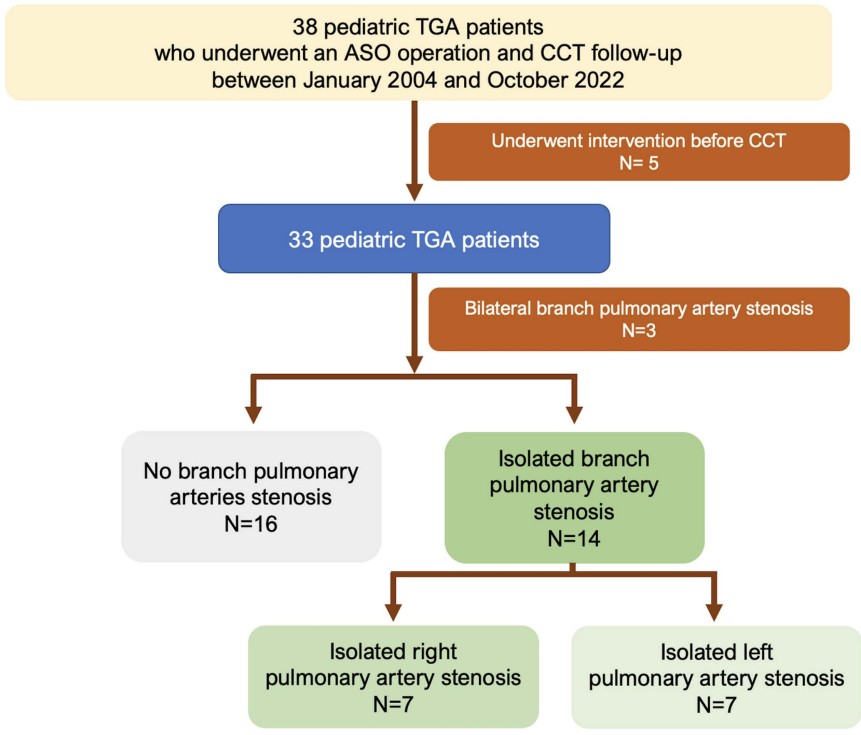

**Fig 1. Flow chart of the study cohort. Abbreviation:** ASO: arterial switch operation; CCT: cardiac computerized tomography; PA: pulmonary artery; TGA: transposition of great artery.

intersect both the anterior border convexity and posterior border of the spinal canal. In cases of vertebral deformity, this line is drawn using the average position between these anatomical markers.

### Data collection

Demographic and clinical data were manually extracted by reviewing the electronic health records. All echocardiograms were performed using a Philips CX 50 and Phillips Epiq 7 with 8 or 12 Mhz probes (Philips Medical Systems) according to patient size. Echocardiography measurements before ASO surgery (First echocardiography data) included pulmonic valve (PV) annulus size, PV annulus z score, PV/BSA, aortic root, aortic root z score, aortic root/ BSA, right pulmonary artery (RPA) size, left pulmonary artery (LPA) size, RPA stenosis, and LPA stenosis. Echocardiography measurements were taken at the time point nearest to the CCT within a six-month period and included age at echocardiography, right atrium enlargement (RAE), right ventricular hypertrophy (RVH), neo-pulmonic valve insufficiency, neo-aortic valve insufficiency, neo-aortic root size, neo-aortic root size/BSA, neo-aortic root z score, left ventricular ejection fraction (LVEF), isolated RPA stenosis, and isolated LPA stenosis. The right atrial enlargement (RAE) and right ventricular enlargement (RVE) were evaluated by echocardiogram using previously published methods [12,13]. Pulsed wave and continuous wave Doppler echocardiography assessed the stenosis of pulmonary artery side branches, in which significant stenosis is obvious where there are measurable gradients of > 20–30 mmHg across the stenosis area [14].

CCT data after ASO included date of CCT, age at time of CCT, duration after ASO surgery, neo-aortic root size, neo-aortic root size/BSA, neo-aortic root z score, ascending aorta diameter, ascending aorta diameter/BSA, ascending aorta diameter z score isolated RPA stenosis, isolated LPA stenosis, bilateral PA stenosis, neo-pulmonary to neo-aortic angle degree, and pulmonary artery bifurcation to AAO angle degree. Z-scores for cardiac CT measurements were

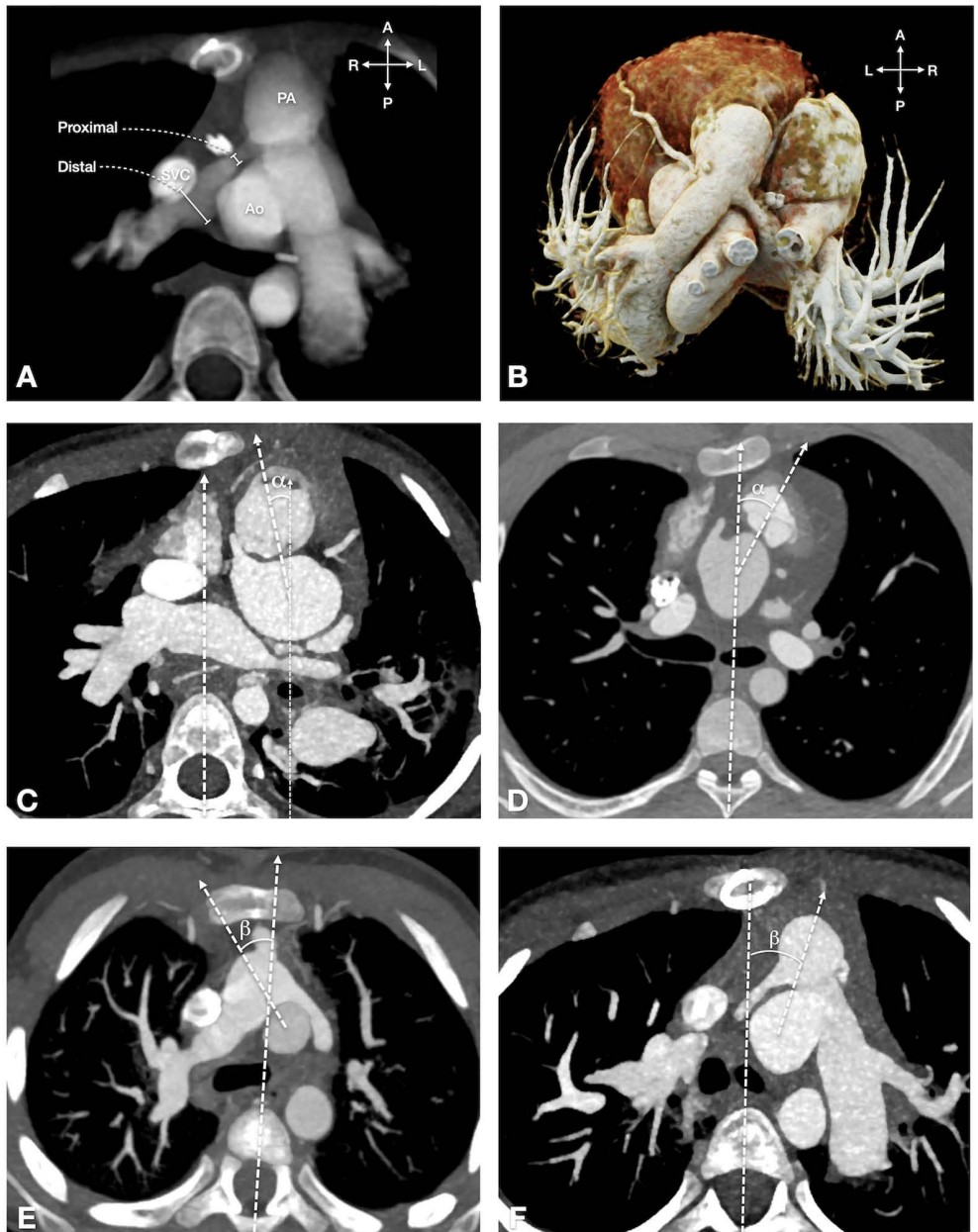

**Fig 2. Great vessel anatomy and angular measurements following the LeCompte Maneuver.** 2A: Great vessel anatomy in an axial plane. Right pulmonary artery stenosis is defined when the ratio of proximal to distal measurements is < 0.75. 2B: Volume-rendered 3D reconstructions of a TGA patient after ASO with LeCompte maneuver, demonstrating RPA stenosis. 2C: Neo-pulmonary to neo-aortic angle (alpha angle). This angle is measured between a line through the centers of the neo-aortic and neo-pulmonary roots and a reference vertical line. For graphic demonstration, a parallel line to the reference vertical line is drawn. A rightward deviation results in a negative alpha angle. 2D: Neo-pulmonary to neo-aortic angle (alpha angle). This angle is measured between a line through the centers of the neo-aortic and neo-pulmonary roots and a reference vertical line. A leftward deviation results in a positive alpha angle. 2E: Pulmonary artery bifurcation to ascending aortic angle (beta angle). This angle is defined between a reference vertical line and a line from the pulmonary artery bifurcation to the center of the ascending aorta (AAO). A rightward deviation results in a negative beta angle. 2F: Pulmonary artery bifurcation to ascending aortic angle (beta angle). This angle is defined between a reference vertical line and a line from the pulmonary artery bifurcation to the center of the ascending aorta (AAO). A leftward deviation results in a positive beta angle. **Abbreviation:** ASO: arterial switch operation: PA: pulmonary artery; RPA right pulmonary artery stenosis.

calculated using published CT standard values for the pulmonary artery branch [15]. All data were retrieved from electronic medical records and registered in RedCap (Vanderbilt University, Nashville, Tennessee).

The inter-rater reliability was assessed using measurements of two independent investigators (KS—cardiac imaging cardiologist, TS—cardiac imaging radiologist) who were blinded to each other's results. The measurement was performed according to the above-described methodology.

**Definition.** To determine pulmonary arterial branch stenosis, the narrowest diameter was compared with the distal part of the bilateral pulmonary arterial branches before branching. Pulmonary artery branches were considered stenotic when the proximal to distal measurements ratio was < 0.75. Pulmonary artery branches were classified as hypoplastic if the Z-score of both proximal and distal parts was < −2 [16].

Balloon atrial septostomy is a catheter-based procedure in which a wire is threaded through the atrial septum. A balloon is inflated in the left atrium and pulled back into the right atrium, creating an opening to enhance blood mixing [3].

## Statistical analysis

All statistical analyses were performed using Stata 17 (StataCorp, College Station, Texas, USA). Clinical data and investigation findings were described as frequency and percentage for categorical variables, and as means with standard deviations (SD) or medians with interquartile ranges (IQR) for continuous variables, depending on the distribution. Differences between patients with and without PA stenosis were assessed using Fisher's exact probability test for categorical variables, and either the independent t-test or Mann-Whitney U test for continuous variables as appropriate.

We evaluated the inter-rater reliability of the neo-pulmonary to neo-aortic angle and the pulmonary artery bifurcation to AAO angle using the intraclass correlation coefficient (ICC). An ICC value below 0.5 indicates poor reliability, between 0.5 and 0.75 indicates moderate reliability, between 0.75 and 0.9 indicates good reliability, and above 0.90 indicates excellent reliability [17]. If both angles showed good or excellent inter-rater reliability, we randomly selected values from one rater for further assessment of the angle properties. In cases where the ICC demonstrates poor reliability, the selection of value between the two raters was determined based on the opinion of a third expert who was not involved in this study. We compared the new angles across three patient groups (Left PA stenosis, Right PA stenosis, and No PA stenosis) using the extended Wilcoxon rank-sum test by Cuzick [18]. The linear trend between the angles and the PA proximal-distal ratio was analyzed using multivariable linear regression adjusted by age, gestational age, birthweight, and sex. Results were considered statistically significant if the p-value was less than 0.05.

## Results

### Study population

Thirty-three patients underwent CCT and were followed up after ASO surgery. Three patients (3/33, 9%) had bilateral branch pulmonary artery stenosis, and sixteen patients (16/33, 48%) had no branch pulmonary artery stenosis. Fourteen patients (14/44, 42%) had either isolated right pulmonary artery (RPA) or isolated left pulmonary artery (LPA) stenosis (Fig 1). The baseline characteristics of the population are summarized in Table 1. Seventeen patients (56.7%) were male. Sixteen patients (53.5%) had TGA with intact ventricular septum. Most of the patients underwent one-stage repair. The ASO was performed at a median age of 12 days (range 8, 25 days) in one-stage operation and 562 days in two-stage operation (defined as pulmonary artery banding and modified Blalock-Taussig shunt before ASO). The common findings of associated structural heart defects were atrial septal defect (ASD), patent ductus arteriosus (PDA), and ventricular septal defect (VSD).

**Isolated branch pulmonary artery stenosis.** Fourteen patients had isolated branch pulmonary artery stenosis. Out of these, 7 (7/14, 50%) resulted in right pulmonary artery stenosis.

**Intraclass correlation coefficient (ICC) for inter-rater reliability.** Average ICC for pulmonary artery bifurcation to ascending aorta angle and neo-pulmonary to neo-aortic angle were 0.989 (95% CI 0.978–0.995) and 0.991 (95% CI 0.978–0.996), respectively. Both angles demonstrated excellent inter-rater reliability.

**Table 1. Demographic and clinical characteristics.**

| Variable | Number of patients, N = 30<br>N (%) |
|---|---|
| Sex | |
| Male | 17 (56.7) |
| Female | 13 (43.3) |
| Birth weight (gram); mean ± SD | 3059.1 ± 484.7 |
| TGA type | |
| TGA IVS | 16 (53.3) |
| TGA VSD | 14 (46.7) |
| Gestational age at delivery (week) | |
| > 37 weeks | 27 (80.0) |
| 34-36 weeks | 2 (6.7) |
| Unknown gestational age | 4 (13.3) |
| Co-morbidity during pregnancy | |
| Maternal GDM | 2 (6.7) |
| Balloon atrial septostomy | 16 (53.3) |
| Type of surgery | |
| Single stage operation (ASO) | 29 (96.7) |
| Two stage operation (PA banding) | 1 (3.3) |
| Age at single stage of ASO (days); median (IQR) | 12 (8, 25) |
| Associate structural heart defect | |
| ASD/PFO | 28 (93.3) |
| PDA | 27 (90.0) |
| VSD | 14 (46.7) |
| Other | 9 (30.0) |

**Abbreviation:** ASD: atrial septum defect; ASO, arterial switch operation; GDM, gestational diabetes; PA: pulmonary artery; PDA: patent ductus arteriosus; PEG1: prostaglandin E1; SD: standard deviation; TGA: transposition of great arteries; IVS: intact ventricular septum; VSD: ventricular septal defect.

**Comparison before ASO of patients with and without isolated branch pulmonary artery stenosis.** Patients' characteristics and anatomy measurements by CCT before ASO operation are summarized in Table 2. Comparing patients with isolated branch PA stenosis with no branch PA stenosis, there were differences in birth weight and gestational age. Cardiac anatomy before ASO operation (pulmonic valve size, aortic root size, right and left pulmonary size) and history of branch PA stenosis before ASO operation showed no significant differences between the two groups of patients.

**Comparison after ASO of patients with and without isolated branch pulmonary artery stenosis.** The median age of patients who underwent echocardiography in the period closest to CCT is 6.1 years old (range 2.2–18.5). The median age of patients who underwent CCT after an ASO operation is 6.5 years old (range 2.2–18.5). The variables of the clinical, echocardiography, and CCT data with and without isolated branch pulmonary artery stenosis are summarized in Table 3. Only one patient with isolated branch pulmonary artery stenosis developed clinical dyspnea. There were no differences in age at the time of echocardiography between the two groups of patients. Right atrium enlargement and right ventricular enlargement is found in both groups. There is no significant difference in the incidence of neo-pulmonic and neo-aortic valve insufficiency between the two patient groups. The neo-aortic root size is not significantly different between the two groups. In this study, echocardiography detected the pulmonary artery branch stenosis in only 50% of the cases.

**Table 2. Baseline clinical and echo finding comparison between patients with and without isolated branch pulmonary artery stenosis before ASO.**

| Variable | Right PA branch stenosis N = 7 | Left PA branch stenosis N = 7 | Isolated PA branch stenosis N = 14 | No PA branch stenosis N = 16 | P-value[§] |
|---|---|---|---|---|---|
| **Clinical finding** | | | | | |
| Sex, male, n (%) | 5 (71.4) | 2 (28.6) | 7 (50.0) | 10 (62.5) | 0.713 |
| Birth weight (gram); mean ± SD | 2438.8 ± 486.3 | 3203.3 ± 502.1 | 2821.1 ± 617.6 | 3249.3 ± 221.8 | 0.019 |
| Gestational age at delivery (week); mean ± SD | 36 ± 2.3 | 36.2 ± 4.7 | 36.1 ± 3.5 | 38.8 ± 0.9 | 0.019 |
| Balloon atrial septostomy | 4 (57.1) | 3 (42.9) | 7 (50.0) | 9 (56.3) | 1.000 |
| Age at ASO (single stage operation), days, median (range) | 10 (6 −147) | 12 (7-20) | 12 (6-147) | 14 (6-66) | 0.441 |
| **Echo finding** | | | | | |
| Pulmonic valve size (cm); mean ± SD | 1.06 ± 0.17 | 1.14 ± 0.20 | 1.09 ± 0.17 | 1.16 ± 0.22 | 0.502 |
| Pulmonic valve size/BSA (cm/m$^2$); mean ± SD | 52.52 ± 14.96 | 41.05 ± 0.47 | 47.62 ± 12.31 | 38.20 ± 9.96 | 0.180 |
| Pulmonic valve z score; median, (IQR) | 2.00 (−0.57, 2.82) | −1.34 (−2.17, −0.51) | −0.51 (−0.57, 2.00) | −0.03 (−1.31, 1.11) | 0.584 |
| Aortic root size (cm/m$^2$); mean ± SD | 1.21 ± 0.11 | 1.06 ± 0.14 | 1.15 ± 0.14 | 1.18 ± 0.09 | 0.562 |
| Aortic root size/BSA (cm/m$^2$); mean ± SD | 47.94 ± 15.28 | 46.14 ± 6.72 | 47.22 ± 11.36 | 25.05 ± 11.98 | 0.017 |
| Aortic root z score; median (IQR) | −0.39 (−0.77, 3.09) | 0.13 (−0.29, 0.55) | −0.29 (−0.39, 0.55) | −1.1 (−1.28, −0.84) | 0.076 |
| RPA stenosis (%) | 1(14.3) | 0 (0) | 1 (7.1) | 0 (0) | 0.257 |
| LPA stenosis (%) | 0 (0) | 0 (0) | 0 (0) | 0 (0) | 0.442 |
| RPA size (cm); mean ± SD | 0.48 ± 0.18 | 0.59 ± 0.13 | 0.52 ± 0.16 | 0.64 ± 0.14 | 0.105 |
| LPA size (cm); mean ± SD | 0.51 ± 0.10 | 0.48 ± 0.11 | 0.50 ± 0.10 | 0.53 ± 0.21 | 0.674 |

[§]: p-value for comparison between any PA stenosis and Absence stenosis;

**Abbreviation:** ASD: atrial septum defect; ASO, arterial switch operation; GDM, gestational diabetes; PA: pulmonary artery; PDA: patent ductus arteriosus; PEG1: prostaglandin E1; SD: standard deviation; TGA: transposition of great arteries; IVS: intact ventricular septum; VSD: ventricular septal defect.

The duration of CCT follow-up after ASO is 6.4 years (range 2.2–18.5). The size of the neo-aortic root and ascending aorta diameter at pulmonary artery bifurcation show no statistically significant differences between the two groups of patients. In the "isolated RPA or LPA stenosis" group, the A-P angle is not statistically different from the no-branch PA stenosis group of patients (p = 0.472). In the "isolated RPA stenosis" group, the angle between the pulmonary artery bifurcation and the AAO had greater leftward rotation than the "no stenosis" group (angle: 25(24,30) VS −2 (−8,65)). In the "isolated LPA stenosis" group, the angle between the pulmonary artery bifurcation and the AAO had greater rightward rotation than the "no stenosis" group (angle −22(−32, −20) VS −2 (−8,6.5)) (Fig 3). The authors observed that no patient had isolated branch pulmonary stenosis, with the angle between the pulmonary artery bifurcation and the AAO between −20 and 20 degrees (Fig 3).

## Discussion

After the successful arterial switch operation, the common long-term sequelae after ASO at follow-up is the branch PA stenosis. In our study, the incidence of branch PA stenosis is 51%, including bilateral and isolated branch pulmonary artery stenosis. The incidence is in alignment with prior studies. Following the LeCompte operation, two-thirds of patients develop PA stenosis, making it the most frequent reason for reintervention [6,19]. Our study found that right and left-branch pulmonary stenosis incidence is equal. These findings differed from a prior study, which showed that stenosis more commonly affected the left pulmonary artery [6]. Lower body weight during surgery has been connected to an

**Table 3. Baseline clinical, EKG, echo, and cardiac CT finding comparison between patients with and without pulmonary artery branch stenosis after ASO.**

| Variable | Right PA branch stenosis N = 7 | Left PA branch stenosis N = 7 | Isolated PA branch stenosis N = 14 | No PA branch stenosis N = 16 | P-value§ |
|---|---|---|---|---|---|
| **Clinical finding** | | | | | |
| Age (year); median (IQR) | 8.1 (6.2, 9.2) | 5.5 (5.1, 13.0) | 7.5 (5.6, 9.2) | 6.3 (4.3, 10.7) | 0.428 |
| Duration after surgery (year); mean (IQR) | 8.1 (6.1, 9.2) | 5.5 (5.0, 13.0) | 7.1 (5.5, 9.7) | 6.3 (4.3, 10.6) | 0.471 |
| Dyspnea | 0 (0) | 1 (14.3) | 1 (7.1) | 0 (0) | 1.000 |
| Chest pain | 0 (0) | 0 (0) | 0 (0) | 0 (0) | NA |
| Palpitation | 0 (0) | 0 (0) | 0 (0) | 0 (0) | NA |
| **Echo finding closed to CCT** | | | | | |
| Age at echocardiography (year); median (IQR) | 7.7 (6.2, 10.6) | 5.2 (4.8, 12.7) | 7.5 (5.6, 10.6) | 5.7 (4.3, 10.7) | 0.231 |
| Right atrium enlargement, n (%) | 2 (28.6) | 2 (28.6) | 4 (28.6) | 3 (18.8) | 0.675 |
| Right ventricle hypertrophy, n (%) | 2 (28.6) | 2 (28.6) | 4 (28.6) | 3 (18.8) | 0.675 |
| Neo-pulmonic valve insufficiency, n (%) | 1 (14.3) | 2 (28.6) | 3 (21.4) | 3 (18.8) | 1.000 |
| Neo-aortic valve insufficiency, n (%) | 4 (57.1) | 3 (42.9) | 7 (50.0) | 5 (31.3) | 0.457 |
| Neo-aortic root size (mm); mean ± SD | 26.4±6.1 | 21.7±6.0 | 24.0±6.3 | 25.3±6.1 | 0.591 |
| Neo-aortic root size/BSA (mm/m2); mean ± SD | 29.5±6.4 | 23.1±11.1 | 26.3±9.3 | 27.3±7.3 | 0.753 |
| Neo-aortic root z score; median (IQR) | 3.11 (1.27, 3.77) | 0.56 (−0.36, 3.51) | 1.99 (0.56, 3.76) | 2.39 (1.38, 3.05) | 0.901 |
| LVEF; mean ± SD | 64.9±5.0 | 65.1±6.4 | 65.0±5.5 | 67.5±5.7 | 0.240 |
| Isolated Right PA stenosis, n (%) | 3 (42.9) | 1 (14.3) | 4 (28.6) | 1 (12.5) | 0.017 |
| Isolated Left PA stenosis, n (%) | 0 (0) | 3 (42.9) | 3 (21.4) | 0 (0) | 0.09 |
| **Cardiac CT finding** | | | | | |
| Age at cardiac CT (year); median (IQR) | 7.8 (6.2, 9.2) | 5.5 (5.1, 13.0) | 7.5 (5.7, 9.2) | 6.3 (4.3, 10.7) | 0.449 |
| Duration after ASO surgery (year); median (IQR) | 7.7 (6.1, 9.0) | 5.5 (5.0, 13.0) | 7.1 (5.5, 9.0) | 6.3 (4.3, 10.6) | 0.494 |
| Neo-aortic root (cm); mean ± SD | 2.80±0.49 | 2.76±0.59 | 2.78±0.52 | 2.83±0.59 | 0.804 |
| Neo-aortic root/BSA (cm/m$^2$); mean ± SD | 3.16±0.75 | 3.25±0.96 | 3.21±0.83 | 3.07±0.83 | 0.669 |
| Neo-aortic sinus z score; median (IQR) | 4.59 (3.86, 5.33) | 3.90 (1.97, 5.69) | 4.35 (3.43, 5.64) | 3.61 (2.49, 5.04) | 0.406 |
| Ascending aorta diameter at pulmonary artery bifurcation (mm); mean ± SD | 1.81±0.34 | 1.57±0.46 | 1.69±0.42 | 1.87±0.38 | 0.239 |
| Ascending aorta diameter at pulmonary artery bifurcation/BSA (cm/m$^2$); mean ± SD | 2.11±0.79 | 1.84±0.58 | 1.98±0.68 | 2.02±0.53 | 0.829 |
| Ascending aorta diameter at pulmonary artery bifurcation, z score; median (IQR) | 0.27 (−1.95, 2.25) | −1.17 (−1.88, 0.28) | −0.26 (−1.88, 0.56) | 0.23 (−0.35, 1.23) | 0.299 |
| Neo-pulmonary to neo-aortic angle (degree); median (IQR) | −10 (−22.5, 9.5) | −14 (−24, −13) | – | −10 (−39, 25) | **0.472*** |
| Pulmonary artery bifurcation to AAO angle (degree); median (IQR) | 25 (24,30) | −22 (−32, −20) | – | −2 (−8,6.5) | **0.042*** |

§: p-value for comparison between any PA stenosis and Absence stenosis; * p-value for comparison across three groups of patients (Left PA stenosis, Right PA stenosis, and No PA stenosis).

**Abbreviation:** AAO: ascending aorta; BSA, body surface area; CT, computer tomography; IQR, interquartile range; PA, pulmonary artery; LVEF, left ventricular ejection fraction.

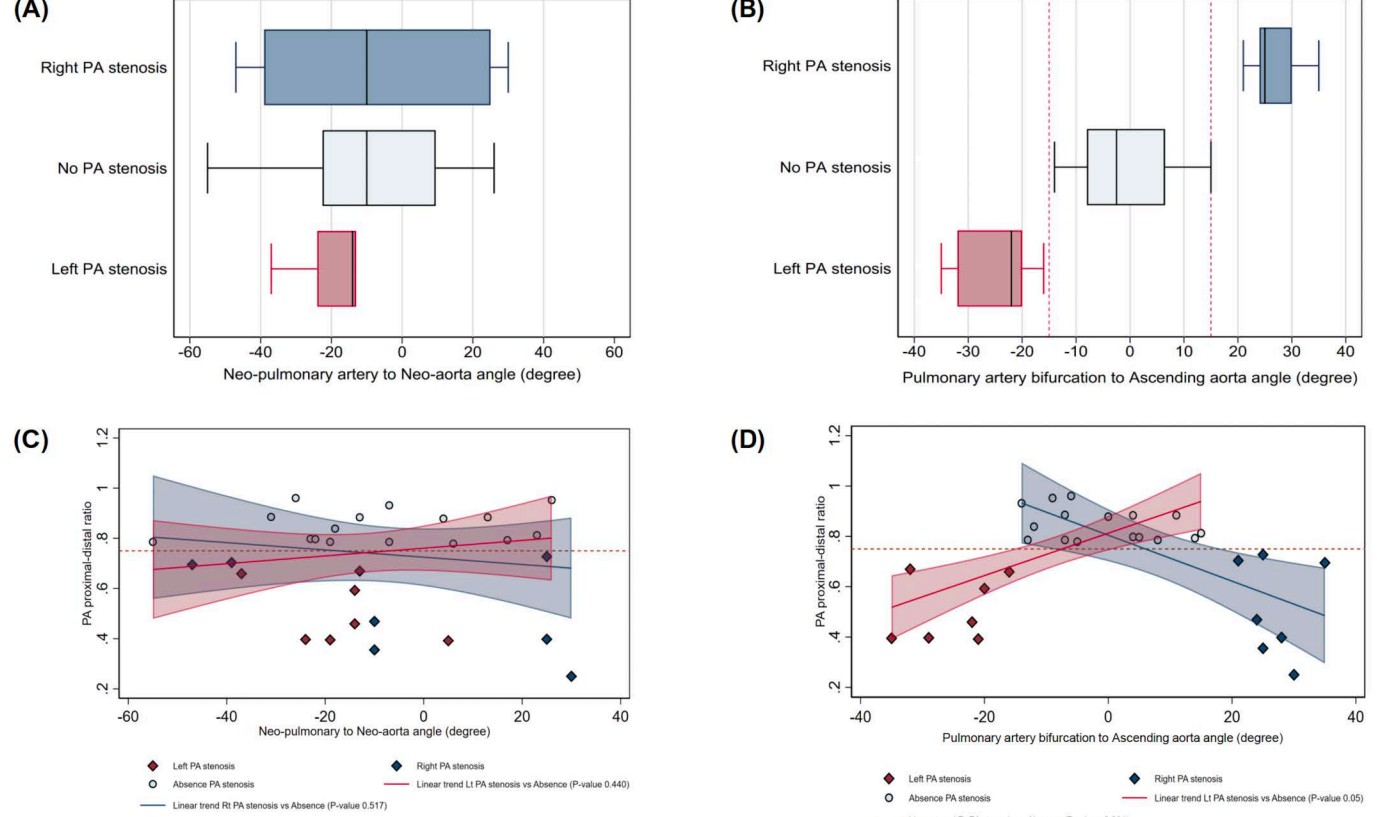

**Fig 3. Angles distribution in isolated branch pulmonary artery stenosis illustrated Box plots (A) Neo-pulmonary artery to neo-aorta angle and (B) Pulmonary artery bifurcation to ascending aorta angle, with linear adjusted trends for PA proximal distal ratio (C, D)** in right PA stenosis (navy), no PA stenosis (bluish grey), and left PA stenosis (red). In **(B)**, no overlap is observed between subgroups. Dashed red lines mark angle thresholds (−15 and 15 degrees), while in **(C, D)**, the horizontal dashed red line indicates a PA stenosis threshold of 0.75. Linear trend P-values for left and right PA stenosis vs no PA stenosis: (C) 0.506 and 0.932; (D) 0.011 and 0.269, respectively. **Abbreviation:** PA: pulmonary artery.

increased risk of obstruction in the pulmonary trunk or branch PA [20]. In our study, the birth weight, close to the time of ASO surgery, was not significantly different between the branch PA stenosis and the no-branch PA stenosis groups. This study also shows that the native anatomy, including pulmonic valve size, aortic root size, and pulmonary branch size, does not show significant differences between the two groups of patients.

Echocardiography detected pulmonary artery stenosis around 50% of the time at follow-up. Echocardiography's limitation in identifying lesions outside the heart could delay intervention or surgery treatment [21]. Doppler measurements of the branch PA were technically unfeasible in 20% of patients from previous studies [22,23]. Broda CR et al. illustrate advanced cardiac imaging methods, e.g., cardiac magnetic resonance (CMR) or CCT, which offer better visualization of the right and left pulmonary arteries than transthoracic echocardiography [24]. With excellent inter-rater reliability, our findings highlight the utility of advanced cardiac imaging, including CCT and CMR, is valuable for evaluating post-ASO patients, especially in patients over five years post-surgery or those with echocardiographic limitations or abnormal heart murmurs. We suggest long-term surveillance of TGA patients following ASO surgery, particularly beyond five years, using advanced cardiac imaging to enable early detection of abnormalities and ensure timely planning for appropriate management, including intervention or surgery.

A previous study attempted to elucidate the mechanisms of obstruction at various levels of the neo-pulmonary outflow tract, including inadequate mobilization of branch pulmonary arteries, reconstruction techniques for the proximal pulmonary artery, posterior compression by the neo-aorta, remnant ductal tissue causing left pulmonary artery stenosis, neo-pulmonary to neo-aortic geometry after ASO, and postoperative compression of the LPA by an enlarged aorta [10].

In our study, neo-pulmonary to neo-aortic geometry after ASO and the size of the aortic root and AAO was not associated with isolated branch PA stenosis but we found that the greater rightward or leftward angulation of the angle between the pulmonary artery bifurcation and the AAO likely contributed to the development of isolated branch PA stenosis after ASO from a possible over-stretching mechanism. The stretched, elongated, and tensioned pulmonary artery course may contribute to the mechanism underlying branch pulmonary stenosis. The proposed explanation for this occurrence is a more abnormal rightward or leftward angle of the PA, which may result in blood from the right ventricle being preferentially diverted to the other side of the PA. This may cause the abnormal velocity profile and wall-shear stress distribution in isolated branch PA stenosis [25]. These findings may indicate the need for refinements in pre-ASO LeCompte surgical techniques, particularly in planning and adjusting the angle between AAO and PA bifurcation, which should be less than 20 degrees.

### Study limitations

The present study has some limitations. First, as an observational retrospective study, it is subject to inherent drawbacks, including data quality issues and potential selection bias. The selection of cases for advanced imaging may have been influenced by echocardiographic limitations, such as technical constraints and suboptimal Doppler ultrasound angles, making pulmonary artery assessment challenging. Additionally, pediatric cardiologists may have selected cases based on heart murmurs, potentially increasing the observed incidence of PA stenosis in our cohort. Second, measurement errors in the angle between the AAO and PA bifurcation may arise due to vertebral deformities (e.g., scoliosis or butterfly vertebrae) and inconsistent selection of the key image for pulmonary artery bifurcation. However, scrolling through image stacks helped identify the most representative bifurcation image. Third, this study was performed at a single center, potentially limiting its generalizability. Lastly, the study population had a small sample size. A multicentre study with a larger patient cohort and a well-standardised long-term surveillance protocol for post-ASO patients using advanced cardiac imaging is warranted.

Future research utilizing 4D flow technology by cardiac MRI for advanced hemodynamic analysis, including wall shear stress and energy loss, may provide new insights into the underlying mechanisms of branch pulmonary artery stenosis. Preoperative planning with computational modeling using computational fluid dynamics may improve surgical strategies to optimize hemodynamics and enhance surgical outcomes.

### Conclusion

The common long-term sequela after ASO is stenosis of the pulmonary artery side branches and its mechanism is not yet determined. An increased rightward or leftward angulation of the pulmonary artery bifurcation and the AAO correlated with isolated branch PA stenosis, possibly resulting from the PA stretched force. Pre-ASO LeCompte strategies to optimize the positioning and angulation of pulmonary artery branches may improve the outcome of surgery. Incorporation of advanced cardiac imaging, e.g., CCT or CMR, into routine surveillance of patients who undergo ASO is warranted.

### Acknowledgments

The authors wish to express gratitude to the significant contributions of Chiang Mai University Hospital in facilitating this research.

### Author contributions

**Conceptualization:** Panupong Seripanu, Tanop Srisuwan, Yupada Pongprot, Rekwan Sittiwangkul, Pakpoom Wongyikul, Phichayut Phinyo, Kwannapas Saengsin.

**Data curation:** Panupong Seripanu, Tanop Srisuwan, Thanaporn Phanacharoensawad.

**Formal analysis:** Panupong Seripanu, Yupada Pongprot, Rekwan Sittiwangkul, Pakpoom Wongyikul, Phichayut Phinyo, Kwannapas Saengsin.

**Investigation:** Thanaporn Phanacharoensawad, Pakpoom Wongyikul, Phichayut Phinyo, Kwannapas Saengsin.

**Supervision:** Yupada Pongprot, Rekwan Sittiwangkul, Kwannapas Saengsin.

**Validation:** Yupada Pongprot, Rekwan Sittiwangkul, Pakpoom Wongyikul, Phichayut Phinyo.

**Writing – original draft:** Panupong Seripanu.

**Writing – review & editing:** Panupong Seripanu, Kwannapas Saengsin.

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
