## [Decision Letter · Decision Letter 0]

Dear Dr. Saengsin,

Thank you for submitting your manuscript to PLOS ONE. After careful consideration, we feel that it has merit but does not fully meet PLOS ONE’s publication criteria as it currently stands. Therefore, we invite you to submit a revised version of the manuscript that addresses the points raised during the review process.

**Please provide point-by-point responses to each reviewer comment in order, without merging any comments. Be concise and specify where you made changes in the manuscript by providing the corresponding page and line numbers.**

We look forward to receiving your revised manuscript.

Kind regards,

Seyedeh Yasamin Parvar, M.D., M.P.H.

Academic Editor

PLOS ONE

**Journal Requirements:**

2. In the online submission form, you indicated that Isolated branch pulmonary artery stenosis after the arterial switch operation in the pediatric transposition of great artery patients does not cover data posting in public databases. However, data are available upon request should be sent to kwannapas_09@hotmail.com and are subject to approval by the Faculty of Medicine, Chiang Mai University Ethics Committee.

Reviewers' comments:

Reviewer's Responses to Questions

**Comments to the Author**

1. Is the manuscript technically sound, and do the data support the conclusions?

Reviewer #1: Yes

Reviewer #2: Yes

Reviewer #3: Yes

Reviewer #4: Partly

2. Has the statistical analysis been performed appropriately and rigorously?

Reviewer #1: Yes

Reviewer #2: Yes

Reviewer #3: Yes

Reviewer #4: I Don't Know

3. Have the authors made all data underlying the findings in their manuscript fully available?

Reviewer #1: No

Reviewer #2: Yes

Reviewer #3: No

Reviewer #4: Yes

4. Is the manuscript presented in an intelligible fashion and written in standard English?

Reviewer #1: Yes

Reviewer #2: Yes

Reviewer #3: Yes

Reviewer #4: Yes

**Reviewer #1:**  Your manuscript is commendable for its clarity and novelty. To enhance its impact, I recommend:

1. Expanding the clinical implications in both the discussion and conclusion sections

2. Improving the visual data representation to ensure clarity and accessibility for readers. (try taking plot exports in a higher quality)

**Reviewer #2:**  Dear Authors,

I found your manuscript interesting, for better illustration, I have some suggestion:

First, it should be better explaining more comprehensive about the way of measurement the angle between AAO and PA bifurcation, also mentioning about the parameters that lead to measurement error.

The second, explaining about two modality imaging(CCT,CMR) difference in evaluation of this angle.

Finally explain your suggestion to the surgeon that how can apply this issue for alleviation PAB stenosis complication.

**Reviewer #3: ** This manuscript provides relevant and compelling data and findings on the clinical significance of the pulmonary artery bifurcation angle. Below are some comments and suggestions that the authors could consider expanding on and clarifying in the paper.

---

Abstract states % without raw numbers.

---

Direct evidence for angulation-stenosis hypothesis?

The introduction could more clearly articulate the importance of the angle of pulmonary artery bifurcation in post-ASO stenosis and anatomical aspects of the pulmonary arteries post-ASO and how these relate to stenosis.

---

The study’s retrospective design introduces selection bias, particularly with exclusions of complex cases. This limits generalizability to the broader TGA population.

Justification for CCT timing could be stronger. The rationale for performing CCT specifically after ASO? Was there a specific clinical context for ordering CCT in these patients, or was it part of a routine follow-up protocol?

---

Briefly explain justification for using the extended Wilcoxon rank-sum test by Cuzick for trend analysis across the three stenosis groups.

30 patients, with only 7 per subgroup (RPA/LPA stenosis), reduce statistical power. Table 3 reports non-significant differences in neo-aortic root size (p=0.591) but may reflect Type II error.

Table 2 reports p=0.019 for gestational age but tests 10+ variables. No correction increases type I error risk.

Table 2 shows a statistical difference in birth weight and gestational age between the groups at baseline. Multivariable analysis to adjust for these factors could better clarify the association between the pulmonary artery bifurcation angle and stenosis.

The manuscript does not clarify how outliers or ambiguous CCT measurements were resolved between raters.

The manuscript states 14 patients had isolated branch PA stenosis, 7 RPA and 7 LPA cases. Table 3 (isolated PA branch stenosis N=14) reports 7 cases, 4 RPA (28.6%) and 3 LPA (21.4%). This leaves 7 unaccounted cases.

---

Does the result section report the calculated ICC values to support 'Both angles demonstrated excellent inter-rater reliability' in the discussion?

Directly mentioning the p-values for the trend analysis in the figure 3 captions would make the figure more clear and self-contained.

Justification for figure 3 '-15/15 degrees' cutoffs?

---

The conclusion suggests 'pre-ASO surgical strategy changes' but lacks evidence that angulation modification improves outcomes.

The 'over-stretching' hypothesis lacks direct evidence. The proposed mechanism as the cause of stenosis due to angle variations lacks strong supporting evidence.

While the study identifies an association between the pulmonary artery bifurcation angle and post-ASO stenosis, the clinical actionability of this finding is not clearly established.

Limitations in design need to be acknowledged in the discussion more explicitly and briefly in the abstract.

Expand challenges and future research directions.

---

**Reviewer #4:**  This manuscript investigates the association between changes in pulmonary artery bifurcation angles and isolated branch pulmonary artery stenosis following the ASO in pediatric patients with TGA. The study provides valuable insights by utilizing cardiac CT imaging to identify geometric alterations potentially contributing to post-operative complications. The findings are significant for improving surgical strategies and post-operative surveillance. However, the manuscript requires improvements in clarity, structure, and depth of discussion to better support its conclusions and enhance reader comprehension. Below are detailed comments for revisions:

1. The title is informative but long.

2. Page 2, Line 10: Define "arterial switch operation" clearly

3. Page 2, Line 33: Expand "The mechanism of branch stenosis is still unclear" by mentioning prior hypotheses or research gaps to highlight the novelty of your study.

4. Page 7, Table 1: Add total percentages for categories such as "sex" and "TGA type" for better interpretation.

5. Page 8, Line 25: Briefly explain "Balloon atrial septostomy" for readers unfamiliar with this procedure.

6. Figure 2: Labeling for angles in Figures 2C and 2F is difficult to follow. Clearer labels or additional descriptions in the legend would improve comprehension.

7. Page 14, Figure 3: Ensure figure legends are detailed enough to be understood without referring back to the text.

8. Page 20, Line 15: Simplify "The proposed explanation for this occurrence is a more abnormal rightward or leftward..."

9. Page 22, Line 3: Expand on how selection bias may have influenced your findings and provide suggestions for mitigating this issue in future studies.

10. Abstract: Consider including more specific results to highlight the study's key findings, such as precise angle measurements.

11. Discussion: The proposed mechanisms of over-stretching should be further elaborated and supported with literature evidence.

**Do you want your identity to be public for this peer review?** For information about this choice, including consent withdrawal, please see our Privacy Policy

Reviewer #1: **Yes: ** Khatere Roozbehi

Reviewer #2: **Yes: ** Zahra Khajali

Reviewer #3: No

Reviewer #4: No

---

## [Author Response · Author response to Decision Letter 1]

26 Feb 2025

The association between a greater rightward or leftward shift of the pulmonary artery bifurcation after the arterial switch operation and contralateral branch pulmonary artery stenosis in the pediatric transposition of great artery patients

Thank you very much for your kind considering of our manuscript. We appreciate the constructive and very helpful feedback and the supportive comments from the reviewers. We have carefully considered the reviewers’ comments and suggestions and have edited the manuscript to address them. Based on those suggestions, we believe the manuscript has greatly improved after the revision.

We have addressed each comment as outlined below with the yellow color highlight in the manuscript.

Reviewer #1: Your manuscript is commendable for its clarity and novelty. To enhance its impact, I recommend:

1.Expanding the clinical implications in both the discussion and conclusion sections

Answer: We would like to thank the reviewer for this suggestion. We expanded the clinical implication in both the discussion and the conclusion sections.

Discussion section

These findings may indicate the need for refinements in pre-ASO LeCompte surgical techniques, particularly in planning and adjusting the angle between AAO and PA bifurcation to less than 20 degrees.

Conclusion section

Pre-ASO LeCompte strategies to optimize the positioning and angulation of pulmonary artery branches may improve the outcome of surgery.

Discussion section: Pages 22, lines 352-354

Conclusion section: Pages 23, lines 379-380

2. Improving the visual data representation to ensure clarity and accessibility for readers. (try taking plot exports in a higher quality)

Answer: We would like to thank the reviewer for this suggestion. We replaced a figure with higher-quality images.

The revised Figure 3 are shown below.

Reviewer #2: Dear Authors,

I found your manuscript interesting, for better illustration, I have some suggestion:

1.it should be better explain more comprehensively the way of measuring the angle between AAO and PA bifurcation, and also mention the parameters that lead to measurement error.

Answer: We would like to thank the reviewer for this suggestion. We explain how to measure the angle between AAO and PA bifurcation. We mention the parameters of the measurement of the angle between AAO an PA bifurcation that can lead to measurement error in the limitation section.

Material and methods section

“Pulmonary artery bifurcation to ascending aortic angle is defined as the angle between two lines: one line extends from the center of the ascending aorta to the pulmonary artery bifurcation, while the other is a reference line drawn from the spinous process to the anterior spinal border convexity. To measure this angle, the axial images are first scrolled through to locate the slice showing the tip of the reverse V-shaped pulmonary arterial bifurcation. A line is then drawn from the tip of the pulmonary arterial bifurcation to the center of the ascending aorta. The reference vertical line is drawn from the tip of the spinous process to the anterior most point of the spinal body convexity. This reference line should pass through the center of the spinal canal and intersect both the anterior border convexity and posterior border of the spinal canal. In cases of vertebral deformity, this line is drawn using the average position between these anatomical markers.”

Study limitation

“Second, measurement errors in the angle between the AAO and PA bifurcation may arise due to vertebral deformities (e.g., scoliosis or butterfly vertebrae) and inconsistent selection of the key image for pulmonary artery bifurcation. However, scrolling through image stacks helps identify the most representative bifurcation image.”

Material and methods section: Page 8, line2 148-158

Study limitation: Page 22 lines 362-366

2.explaining about two modality imaging (CCT, CMR) difference in evaluation of this angle.

Answer: In considering the differences between CCT and CMR, these angles rely on bony landmarks for measurement. With CMR imaging, obtaining an accurate vertical reference line may be challenging because bony landmarks may not be as well visualized as on the CCT due to the inherent characteristics of MR imaging.

3.explain your suggestion to the surgeon that how can apply this issue for alleviation PAB stenosis complication.

Answer: We would like to thank the reviewer for this suggestion. We suggest to the surgeon how to apply this issue for alleviation of branch pulmonary arteries stenosis complication

“These findings may indicate the need for refinements in pre-ASO LeCompte surgical techniques, particularly in planning and adjusting the angle between AAO and PA bifurcation to less than 20 degrees.”

Page22 lines 352-354

Reviewer #3: This manuscript provides relevant and compelling data and findings on the clinical significance of the pulmonary artery bifurcation angle. Below are some comments and suggestions that the authors could consider expanding on and clarifying in the paper.

1. Abstract states % without raw numbers.

Answer: We would like to thank the reviewer for this suggestion. We add raw data number in the abstract.

Of the thirty patients enrolled in this study, 46.67% (14/30) had an isolated branch pulmonary artery stenosis, and 53.33% (16/30) had no pulmonary artery stenosis.

Page2 lines 34-36

2. Direct evidence for angulation-stenosis hypothesis?

Answer: We would like to thank the reviewer for this question. To the best of our knowledge, there is no publication showing the correlation of pulmonary artery bifurcation to ascending aortic angle and isolated branch PA stenosis. Morgan CT et al. showed that neo-pulmonary to neo-aortic geometry impacts LPA size and perfusion of the left lung. Allen H et.al. showed that the branch pulmonary stenosis likely arises from multiple causes, with post-ASO obstruction potentially resulting from the pulmonary elongated and stretched pathway.

References

1. Morgan CT, Mertens L, Grotenhuis H, Yoo SJ, Seed M, Grosse-Wortmann L. Understanding the mechanism for branch pulmonary artery stenosis after the arterial switch operation for transposition of the great arteries. Eur Heart J Cardiovasc Imaging. 2017 Feb;18(2):180-185. doi: 10.1093/ehjci/jew046. Epub 2016 Mar 29. PMID: 27025515.

2. Allen H, Driscoll DJ, Shaddy RE, Feltes TF. Moss and Adams Heart Disease in Infants, Children, and Adolescents: Including the Fetus and Young Adult. 8th ed. Philadelphia: Wolters Kluwer Health/Lippincott Williams & Wilkins; 2013. p1792.

3. The introduction could more clearly articulate the importance of the angle of pulmonary artery bifurcation in post-ASO stenosis and anatomical aspects of the pulmonary arteries post-ASO and how these relate to stenosis.

Answer: We would like to thank the reviewer for this suggestion. In the introduction, we added the important hypothesis in post-ASO stenosis and anatomical aspects of the pulmonary arteries post-ASO.

“According to a previous study, geometric anatomical factors contributing to branch pulmonary stenosis likely have multiple causes, with post-ASO branch pulmonary stenosis potentially arising from a lengthened and tensioned pulmonary artery pathway”

“We hypothesize the new geometric anatomy orientations, including the angle between AAO and PA bifurcation after surgery are the risk factors of branch PA stenosis.”

Page 4, lines 68-70

Page 5, lines 78-79

4. The study’s retrospective design introduces selection bias, particularly with exclusions of complex cases. This limits generalizability to the broader TGA population.

Justification for CCT timing could be stronger. The rationale for performing CCT specifically after ASO? Was there a specific clinical context for ordering CCT in these patients, or was it part of a routine follow-up protocol?

Answer: We would like to thank the reviewer for this suggestion. Due to the retrospective study, we didn’t have a protocol or routine for follow-up post-op ASO at that time. We explored the specific clinical conditions for ordering CCT in more detail. We found that the median age of CCT was 7.8 years old. The clinical condition for ordering CCT was the limitations of echocardiographic imaging that made it difficult to assess the pulmonary arteries conclusively; obtaining branch PA Doppler measurements was technically unfeasible because of echocardiographic limitations and an improper Doppler ultrasound angle. Pediatric cardiologists suspected branch PA stenosis based on the presence of a heart murmur. We discuss this in more detail in the discussion section.

“With excellent inter-rater reliability, our findings highlight the utility of advanced cardiac imaging, including CCT and CMR, is valuable for evaluating post-ASO patients, especially in patients over five years post-surgery or those with echocardiographic limitations or abnormal heart murmurs. We suggest long-term surveillance of TGA patients following ASO surgery, particularly beyond five years, using advanced cardiac imaging to enable early detection of abnormalities and ensure timely planning for appropriate management, including intervention or surgery.”

Page 21, lines 332-337

5. Briefly explain justification for using the extended Wilcoxon rank-sum test by Cuzick for trend analysis across the three stenosis groups.

Answer: Thank you for your feedback. In certain situations, slope estimates, along with their corresponding confidence intervals, can provide more meaningful insights than hypothesis tests assessing a zero slope [1]. We hypothesise that there is an ordered relationship between the new angle and the categories of Left PA stenosis, No PA stenosis, and Right PA stenosis.

An extension of the Wilcoxon rank-sum trend test has been developed to accommodate data where a variable is measured across three or more ordered groups. Furthermore, this trend test demonstrates greater statistical power as it captures a more precise confidence interval [2].

Reference

1. Lui KJ, Cumberland WG. A Wilcoxon-type test for trend. Stat Med. 1995;14(4):445-446. doi:10.1002/sim.4780140409

2. Cuzick J. A Wilcoxon-type test for trend. Stat Med. 1985;4(1):87-90.

6. 30 patients, with only 7 per subgroup (RPA/LPA stenosis), reduce statistical power. Table 3 reports non-significant differences in neo-aortic root size (p=0.591) but may reflect Type II error.

Answer: Thank you for your feedback. The statistical analysis was performed between the isolated stenosis group (N = 14) and the non-stenosis group (N = 16). While we recognise the small sample size, this limitation has been acknowledged.

“Lastly, the study population had a small sample size. A multicentre study with a larger patient cohort and a well-standardised long-term surveillance protocol for post-ASO patients using advanced cardiac imaging is warranted.”

Page 2, lines 367-369

7. Table 2 reports p=0.019 for gestational age but tests 10+ variables. No correction increases type I error.

8. Table 2 shows a statistical difference in birth weight and gestational age between the groups at baseline. Multivariable analysis to adjust for these factors could better clarify the association between the pulmonary artery bifurcation angle and stenosis.

Answers for no7-8: We appreciate the opportunity to address your concerns. We agree that without correction, the probability of a statistically significant outcome occurring by chance is increased. However, the primary objective of our study is to explore the possible mechanisms of branch PA stenosis using CCT parameters. Therefore, adjusting the P-value for these factors may not be a primary concern in this context.

Regardless of whether the correction leads to rejecting the null hypothesis or not, as you have suggested, multivariable analysis adjusted for gestational age, birth weight, or sex should be conducted when there is a biologically plausible.

We have edited the method accordingly in line 215-217:

“The linear trend between the angles and the PA proximal-distal ratio was analyzed using multivariable linear regression adjusted by age gestational age birthweight and sex.”

Page 11, line 215-217

The revised Figure 3 are shown below.

The association between the pulmonary artery bifurcation-to-ascending aorta angle in patients with right PA stenosis and those without PA stenosis has changed to a minimal effect and is no longer statistically significant. All other associations remain consistent with the unadjusted trend.

9. The manuscript does not clarify how outliers or ambiguous CCT measurements were resolved between raters.

Answer: Thank you for your feedback. We apologise for not providing more details on this issue. In cases where the ICC demonstrates poor reliability, the selection of value between the two raters will be determined based on the opinion of a third expert who was not involved in this study.

We have added the method accordingly in line 210-213: “If both angles showed good or excellent inter-rater reliability, we randomly selected values from one rater for further assessment of the angle properties. In cases where the ICC demonstrates poor reliability, the selection of value between the two raters will be determined based on the opinion of a third expert who was not involved in this study.”

Page 11, lines 210-213

10. The manuscript states 14 patients had isolated branch PA stenosis, 7 RPA and 7 LPA cases. Table 3 (isolated PA branch stenosis N=14) reports 7 cases, 4 RPA (28.6%) and 3 LPA (21.4%). This leaves 7 unaccounted cases.

Answer: Thank you for your feedback. We apologized for the confusion in our manuscript. The mentioned results aim to highlight inaccuracies in echocardiographic findings. Approximately 50% of patients with isolated PA stenosis were underdiagnosed by echocardiography. We have stated this finding in lines 277-278 as follow:

“In this study, echocardiography detected the pulmonary artery branch stenosis in only 50% of the cases.”

And in the discussion in lines 326-329 as follows:

“Echocardiography detected pulmonary artery stenosis around 50% of the time at follow-up. Echocardiography's limitation in identifying lesions outside the heart could delay intervention or surgery treatment (20). Doppler measurements of the branch PA were technically unfeasible in 20% of patients from previous studies (21, 22).”

Result section, Page 15, lines 277-278

Discussion section, Pages 20-21, line326-329

11. Does the result section report the calculated ICC values to support 'Both angles demonstrated excellent inter-rater reliability' in the discussion?

Answer: Thank you for your suggestions. We have added a discussion on this issue in line 336-342 as follow “With excellent inter-rater reliability, our findings highlight the utility of advanced cardiac imaging, including CCT and CMR, is valuable for evaluating post-ASO patients, especially in patients over five years post-surgery or those with echocardiographic limitations or abnormal heart murmurs. We suggest long-term surveillance of TGA patients following ASO surgery, particularly beyond five years, using advanced cardiac imaging to enable early detection of abnormalities and ensure timely planning for appropriate management, including intervention or surgery.”

12. Directly mentioning the p-values for the trend analysis in the figure 3 captions would make the figure clearer and self-contained.

Answer: Thank you for your suggestions. We have provided the p-value on each linear adjusted trend within the legend and figure caption as shown below.

“Fig 3. Angles Distribution in Isolated Branch Pulmonary Artery Stenosis illustrates a Box plots (A) Neo-pulmonary artery to neo-aorta angle and (B) Pulmonary artery bifurcation to ascending aorta angle, with linear adjusted trends for PA proximal distal ratio (C, D) in right PA stenosis (navy), no PA stenosis (bluish grey), and left PA stenosis (red). In (B), no overlap is observed between subgroups. Dashed red lines mark angle thresholds (-15 and 15 degrees), while in (C, D), the horizontal dashed red line indicates a PA stenosis threshold of 0.75. Linear trend P-values for left and right PA stenosis vs no PA stenosis: (C) 0.506 and 0.932; (D) 0.011 and 0.269, resp

---

## [Decision Letter · Decision Letter 1]

Dear Dr. Saengsin,

We look forward to receiving your revised manuscript.

Kind regards,

Seyedeh Yasamin Parvar, M.D., M.P.H.

Academic Editor

PLOS ONE

Journal Requirements:

Additional Editor Comments (if provided):

Congratulations on your excellent paper. Consider adding an abbreviation table at the beginning of the manuscript for clarity. The abstract should follow a structured format, and the results section should be improved to better represent the study's findings. In line 53, the statement “The prevalence of TGA is between 20-30 per 100,000 live births” needs more context—please specify the location and time period. Additionally, extract the approval date from the text and include it in the cover letter or title page.

Reviewers' comments:

Reviewer's Responses to Questions

**Comments to the Author**

Reviewer #1: All comments have been addressed

Reviewer #2: All comments have been addressed

Reviewer #3: All comments have been addressed

Reviewer #4: All comments have been addressed

2. Is the manuscript technically sound, and do the data support the conclusions?

Reviewer #1: Yes

Reviewer #2: Yes

Reviewer #3: Yes

Reviewer #4: Partly

3. Has the statistical analysis been performed appropriately and rigorously?

Reviewer #1: Yes

Reviewer #2: Yes

Reviewer #3: Yes

Reviewer #4: Yes

4. Have the authors made all data underlying the findings in their manuscript fully available?

Reviewer #1: No

Reviewer #2: Yes

Reviewer #3: No

Reviewer #4: Yes

5. Is the manuscript presented in an intelligible fashion and written in standard English?

Reviewer #1: Yes

Reviewer #2: Yes

Reviewer #3: Yes

Reviewer #4: Yes

Reviewer #1: Thank you for addressing the requested revisions. The expanded discussion on clinical implications adds valuable depth to the manuscript, and the improved figure quality significantly enhances clarity and visualization. I appreciate your efforts in strengthening the paper.

Reviewer #2: Thanks for replying and explaining comprehensively to suggestion. All the questions replied and formatted correctly.

Reviewer #3: I suggest that the authors add a clearer and more detailed context to the paper from their replies to comments 4 and 5 .

Reviewer #4: Thank you for addressing all the issues and providing clear explanations. Your revisions improve the clarity and comprehensiveness of the manuscript.

**Do you want your identity to be public for this peer review?** For information about this choice, including consent withdrawal, please see our Privacy Policy

Reviewer #1: **Yes: ** Khatere Roozbehi

Reviewer #2: **Yes: ** Zahra Khajali

Reviewer #3: No

Reviewer #4: No

---

## [Author Response · Author response to Decision Letter 2]

28 Apr 2025

Association of Pulmonary Artery Bifurcation Angle Shift with Contralateral Stenosis after Post-arterial Switch in the Pediatric Transposition of Great Artery Patients

Old Title: The association between a greater rightward or leftward shift of the pulmonary artery bifurcation after the arterial switch operation and contralateral branch pulmonary artery stenosis in the pediatric transposition of great artery patients

Thank you very much for your kind consideration of our manuscript. We appreciate the constructive and very helpful feedback and the supportive comments from the reviewers. We have carefully considered the reviewers’ comments and suggestions and have edited the manuscript to address them. Based on those suggestions, we believe the manuscript has greatly improved after the revision.

Revision 1

We have addressed each comment as outlined below with the yellow color highlight in the manuscript.

Revision 2

We have addressed each comment as outlined below with the green color highlight in the manuscript.

#Additional Editor Comments: Congratulations on your excellent paper. Consider adding an abbreviation table at the beginning of the manuscript for clarity. The abstract should follow a structured format, and the results section should be improved to better represent the study's findings. In line 53, the statement “The prevalence of TGA is between 20-30 per 100,000 live births” needs more context—please specify the location and time period. Additionally, extract the approval date from the text and include it in the cover letter or title page.

Answer:

• Comment 1: Consider adding an abbreviation table at the beginning of the manuscript for clarity.

o Response: Thank you for the suggestion. We have now included an Abbreviation Table at the beginning of the manuscript, listing all relevant terms used for better clarity and ease of reading.

o Page 4

• Comment 2: The abstract should follow a structured format, and the results section should be improved to better represent the study's findings.

o Response: We appreciate this recommendation. The abstract has been revised into a structured format with the following headings: Background, Methods, Results, and Conclusion, in accordance with PLOS ONE guidelines. We also rewrote the result section in the abstract part to better represent the study's findings.

o Page 2-3 lines 26-47

• Comment 3: In line 53, the statement “The prevalence of TGA is between 20-30 per 100,000 live births” needs more context

o Response: Thank you for pointing this out. We have revised the sentence for clarity and context. It now reads:

“The prevalence of TGA is estimated to be between 20 and 30 per 100,000 live births, based on population-based studies conducted in Europe and North America from 2000 to 2010.” A corresponding reference has been added.

o Reference: van der Linde D, Konings EE, Slager MA, et al. Birth prevalence of congenital heart disease worldwide: a systematic review and meta-analysis. J Am Coll Cardiol. 2011;58(21):2241-2247. doi:10.1016/j.jacc.2011.08.025

o Page 5 line 71-73

• Comment 4: Please specify the location and time period. Additionally, extract the approval date from the text and include it in the cover letter or title page.

o Response: The Institutional Review Board of the Faculty of Medicine, Chiang Mai University, Chiang Mai, Thailand, has been extracted from the Methods section and is now included in the cover letter.

o The study was approved by the Institutional Review Board of the Faculty of Medicine, Chiang Mai University, Chiang Mai, Thailand, on 16/11/2022 (approval no. 411/2022, date of approval 16/11/2022, and date of expiration 15/11/2023).

o Page 6-7 lines 111-120

o Cover letter

# Reviewers' comments:

2. Is the manuscript technically sound, and do the data support the conclusions?

Answer: We would like to thank reviewers to this comment. The conclusions are directly supported by the data, particularly the observed relationship between PA bifurcation angles and isolated branch pulmonary artery stenosis and are framed within the context of the study’s limitations.

“Alterations in the PA bifurcation angle relative to the AAO may contribute to the development of isolated branch PA stenosis following ASO. Preoperative planning to optimize the PA bifurcation angle to within ±20 degrees may help reduce the risk of postoperative stenosis. A larger study incorporating advanced cardiac imaging into routine ASO follow-up is warranted.”

Page 2-3 line 44-47

4. The PLOS Data policy requires authors to make all data underlying the findings described in their manuscript fully available without restriction, with rare exception (please refer to the Data Availability Statement in the manuscript PDF file). The data should be provided as part of the manuscript or its supporting information or deposited to a public repository. For example, in addition to summary statistics, the data points behind means, medians and variance measures should be available. If there are restrictions on publicly sharing data—e.g. participant privacy or use of data from a third party—those must be specified.

Answer: Thank you for your comment regarding data availability.

We confirm that the data underlying the findings of this study are available upon reasonable request. Due to institutional policies and patient privacy considerations, the raw data cannot be publicly shared in a repository. However, we are fully committed to transparency and will provide the individual-level data points, including those behind the reported means, medians, and variance measures, to researchers upon request.

This has been updated in the Data Availability Statement as follows:

Data Availability Statement: The data underlying this study cannot be made publicly available due to patient confidentiality restrictions. However, they will be made available to qualified researchers upon reasonable request. Please contact Kwannapas_09@hotmai.com for access and requests are subject to approval by the Faculty of Medicine, Chiang Mai University Ethics Committee.

Page 25 lines 417-420

---

## [Editor Report · Decision Letter 2]

Association of Pulmonary Artery Bifurcation Angle Shift with Contralateral Stenosis after Post-arterial Switch in the Pediatric Transposition of Great Artery Patients

PONE-D-24-53522R2

Dear Dr. Saengsin,

We’re pleased to inform you that your manuscript has been judged scientifically suitable for publication and will be formally accepted for publication once it meets all outstanding technical requirements.

Kind regards,

Seyedeh Yasamin Parvar, M.D., M.P.H.

Academic Editor

PLOS ONE
---

## [Editor Report · Acceptance letter]

PONE-D-24-53522R2

PLOS ONE

Dear Dr. Saengsin,

I'm pleased to inform you that your manuscript has been deemed suitable for publication in PLOS ONE. Congratulations! Your manuscript is now being handed over to our production team.

Kind regards,

on behalf of

Dr. Seyedeh Yasamin Parvar

Academic Editor

PLOS ONE